# Activation of Soluble Polysaccharides with 1-Cyano-4-Dimethylaminopyridine Tetrafluoroborate (CDAP) for Use in Protein–Polysaccharide Conjugate Vaccines and Immunological Reagents. III Optimization of CDAP Activation

**DOI:** 10.3390/vaccines8040777

**Published:** 2020-12-18

**Authors:** Andrew Lees, Jackson F. Barr, Samson Gebretnsae

**Affiliations:** 1Fina Biosolutions LLC, 9430 Key West Ave, Suite 200, Rockville, MD 20850, USA; JBarr@som.umaryland.edu (J.F.B.); Samson@Finabio.com (S.G.); 2School of Medicine, University of Maryland, Baltimore, MD 21201, USA

**Keywords:** CDAP, CNBr, conjugate vaccine, conjugation, polysaccharide, dextran, 1-cyano-4-dimethylaminopyridine tetrafluoroborate, DMAP

## Abstract

CDAP (1-cyano-4-dimethylaminopyridine tetrafluoroborate) is employed in the synthesis of conjugate vaccines as a cyanylating reagent. In the published method, which used pH 9 activation at 20 °C (*Vaccine*, 14:190, 1996), the rapid reaction made the process difficult to control. Here, we describe optimizing CDAP activation using dextran as a model polysaccharide. CDAP stability and reactivity were determined as a function of time, pH and temperature. While the rate of dextran activation was slower at lower pH and temperature, it was balanced by the increased stability of CDAP, which left more reagent available for reaction. Whereas maximal activation took less than 2.5 min at pH 9 and 20 °C, it took 10–15 min at 0 °C. At pH 7 and 0 °C, the optimal time increased to >3 h to achieve a high level of activation. Many buffers interfered with CDAP activation, but DMAP could be used to preadjust the pH of polysaccharide solutions so that the pH only needed to be maintained. We found that the stability of the activated dextran was relatively independent of pH over the range of pH 1–9, with the level of activation decreased by 40–60% over 2 h. The use of low temperature and a less basic pH, with an optimum reaction time, requires less CDAP, improving activation levels while making the process more reliable and easier to scale up.

## 1. Introduction

The development of conjugate vaccines is one of the great advances in vaccinology. While antibodies to polysaccharides present on bacteria can be protective, immunization with polysaccharides alone generally does not induce an anamnestic immune response, cause class switching or show antibody affinity maturation [1]. Significantly, as T-cell independent antigens, polysaccharides are poorly immunogenic in infants [1]. These shortcomings have been overcome by chemically linking a polysaccharide to a carrier protein to form a conjugate vaccine, as first observed by Goebel and Avery [2,3]. Following the introduction of the *Haemophilus influenzae* type b (Hib) conjugate vaccine, cases of Hib, once one of the most common reasons for pediatric ER visits, became rare [4]. Conjugate vaccines for *Streptococcus pneumoniae* and meningococcal disease are now routinely administered. These vaccines, which are often multivalent, are complex and challenging to manufacture [5].

Most polysaccharides do not have a convenient chemical handle for conjugation, so the polysaccharide must first be made reactive or “activated”. The activated polysaccharide is then linked either directly with the protein (or modified protein) or is functionalized for additional derivatization prior to conjugation (Figure 1).

Most licensed vaccines use either reductive amination or cyanylation to activate polysaccharide hydroxyls. The first conjugate vaccines using cyanylation chemistry employed cyanogen bromide (CNBr) [4,6], a reagent which had previously been used to activate chromatography resins [7,8]. CNBr requires high pH, typically ~pH 10.5 or greater, to partially deprotonate the hydroxyls so they are sufficiently nucleophilic to attack the cyano group. While CNBr is a low-cost and well-established reagent, it is toxic and can be difficult to work with. At the high pH necessary for the hydroxyls to be sufficiently nucleophilic to react with the reagent, neither CNBr nor the cyanate ester intermediate are stable and most of the reagent either hydrolyzes or undergoes nonproductive or poorly reactive side reactions [9]. Furthermore, the high pH required for CNBr activation can hydrolyze or damage alkaline-sensitive polysaccharides.

An improved method of cyanylation of soluble polysaccharides using 1-cyano-4-dimethylaminopyridine tetrafluoroborate (CDAP) was introduced in 1996 by Lees et al. [10]. In contrast to CNBr, CDAP is crystalline and easy to handle. CDAP was found to activate polysaccharides at a lower pH than CNBr and with fewer side reactions [11,12,13]. Furthermore, unlike CNBr-activated polysaccharides, CDAP-activated polysaccharides could be directly conjugated to proteins, simplifying the process as there is no need to isolate a derivatized polysaccharide intermediate. The entire activation and conjugation process can be completed in a single day, in contrast with the multiday process needed with CNBr activation or reductive amination. An additional commercial advantage of direct conjugation was that it avoided chemistry which was under patent in the 1990s [14].

As originally described, CDAP activation of polysaccharide was quite rapid (<3 min), making the process difficult to reproducibly control and challenging to scale up. The protocol used by Lees et al. [10] described adding CDAP from a 100 mg/mL stock to the polysaccharide solution and then raising the pH 30 s later with “an equal volume of 0.2 M triethylamine”. The protein was added after 2.5 min of reaction at room temperature. Notably, the pH of the activation step was not well controlled in the described protocol and most likely initially exceeded the target pH. We noted that sodium hydroxide could be substituted for triethylamine [10], and this base is now generally employed [15].

Conjugation to the activated polysaccharide typically occurs at around pH 9 for several hours, but this pH is not always suitable for pH-sensitive polysaccharides. This limitation was circumvented with the finding that nucleophiles having a lower pKa than primary amines, such as hydrazides (e.g., adipic dihydrazide), were reactive with CDAP-activated polysaccharides over a wide pH range [16]. For example, hydrazide-derivatized proteins could be coupled to CDAP-activated polysaccharides, even at acidic pH [16]. This approach, developed to allow more controlled conjugation, permits using CDAP chemistry for the linking of pH sensitive polysaccharides. However, a somewhat unexpected advantage of using hydrazide-derivatized proteins emerged with the finding of improved coupling efficiency with pH 9 conjugation because of the increased number of reactive groups available (e.g., lysines + hydrazides) (Lees, unpublished data).

### CDAP Conjugation Overview

The CDAP protocol can be conceptualized as two phases: (1) the activation of the polysaccharide and (2) conjugation of the activated polysaccharide with a protein or ligand. The goal of the first step is to efficiently activate the polysaccharide, while the goal of the second is to efficiently conjugate to the activated polysaccharide. The activated polysaccharide ties the two steps together. Figure 2a is an overview of the CDAP activation process, showing the chemical structure of CDAP and the hydrolysis product, DMAP. This conceptualization helps focus on the critical elements of each step. Figure 2b expands on this conceptualization, showing the desired activation and coupling reactions, along with the hydrolysis reactions and some of the side reactions [9]. Although a cyanoester is shown, as reviewed in the Discussion section, this may not be the only reactive intermediate. We therefore refer to the intermediate as (CDAP) “activated” polysaccharide.

During the activation phase, the three major concerns are CDAP stability, CDAP reaction with the polysaccharide hydroxyls and the stability of the activated polysaccharide. CDAP hydrolysis increases with pH, as does the hydrolysis of the activated polysaccharide and the side reactions. However, the CDAP reaction with the polysaccharide is facilitated by increasing the pH. Efficiently activating polysaccharides with CDAP requires a balance between (1) the reactivity of the polysaccharide and CDAP and (2) the hydrolysis and side reactions of both the reagent and the activated polysaccharide.

Despite its widespread use in both research and licensed vaccines, to date, there has been no published study on optimizing CDAP chemistry. At the time of the 1996 publication, the chemistry was not fully understood by the authors and as CDAP is used in commercial vaccines, processes have been kept proprietary. Our laboratory has been working to better understand CDAP chemistry and its use for conjugate vaccine synthesis. This manuscript is an effort to share that knowledge with a focus on CDAP properties and polysaccharide activation. A future publication will address conjugation of proteins to CDAP-activated polysaccharides (manuscript in preparation).

## 2. Materials and Methods

CDAP was originally sourced from Research Organics (Cleveland, OH, USA) which was subsequently acquired by Sigma-Aldrich (St. Louis, MO, USA). The product is still available from Sigma-Aldrich as product #RES1458C, and we recommend its use, as we have found variations in the quality from some other suppliers. Low endotoxin BSA was obtained from SigmaAldrich. Chemicals and reagents were of ACS grade. Dextran T2000 was purchased from Pharmacosmos A/S (Holbaek, Denmark). Other polysaccharides were obtained from the Serum Institute of India (Pune, India) or Inventprise LLC (Redmond, WA, USA). Trinitrobenzene bovine serum albumin (TNP_1_-BSA) was prepared as previously described [17].

Primary amines were assayed with a 2,4,6-Trinitrobenzene sulfonic acid (TNBS) assay as previously described [10], using either adipic dihydrazide (ADH, 2 mole hydrazide per mole ADH) or glycine as standards for hydrazide or amines, respectively. Our TNBS solid was obtained from Kodak Chemicals (Rochester, NY, USA) and is no longer manufactured. A substitute is available from Sigma-Aldrich (#P2297). Dextran was assayed using resorcinol/sulfuric acid [10] with glucose as the standard. CDAP stock solutions were freshly prepared at 100 mg/mL in acetonitrile and diluted to working concentrations in the study buffers or activation reactions.

Aqueous 2.5 M DMAP stock solutions were prepared by adding solid DMAP to deionized water to about two thirds of the final volume. 10 N HCl was carefully added in small increments while mixing until the solution became clear. Small aliquots of 10 N NaOH were then added until the pH was around 8.5 and the solution made up to volume. Final adjustment of the pH is described in the text.

CDAP stability studies were performed by diluting from the 100 mg/mL CDAP stock into designated test solutions kept at room temperature or in an ice water bath. pH was monitored with a Mettler-Toledo (Columbus, OH, USA) SevenMulti meter, equipped with a semi-micro epoxy pH probe and a temperature probe. CDAP hydrolysis was monitored using the spectral difference between CDAP and DMAP, its hydrolysis product, as the absorbance peaks are 301 and 280 nm, respectively, in 0.1 M HCl. Kohn and Wilchek reported an extinction coefficient for CDAP of 28,600 M^−1^ cm^−1^ at 301 nm in 0.1M HCl [18]. DMAP in 0.1 M HCl at 312 nm and 315 nm has about 4% and 3.5%, respectively, of the absorbance of CDAP. CDAP hydrolysis was followed by transferring aliquots from the hydrolysis reaction into 0.1 M HCl. If necessary, further dilutions in acid were made to bring the absorbance into the linear range. The absorbance was read after 30 min. Over the course of this work, we variously used 310–320 nm to monitor CDAP. To determine percent changes, “100%” was considered the absorbance of CDAP immediately diluted in 0.1 M HCl while “0%” was the absorbance of CDAP first transferred to 0.1 M NaOH and then diluted into 0.1 M HCl.

For dextran activation, solutions of 5 or 10 mg/mL were prepared in water or buffered solutions. CDAP was added to rapidly stirred solutions and the pH raised to and/or maintained at the target with 0.1 M NaOH. A Schott Titroline titrator was used to add 0.1 M NaOH in 10 µL doses. Further details and variations are provided in the figure legends. Details for the activation of other polysaccharides are in the figure legends.

For determining the extent of polysaccharide activation, an aliquot of the activation reaction solution was combined with an equal volume of 0.5 M adipic dihydrazide. The reaction was allowed to go to completion and the polysaccharide–hydrazide product was desalted, either by extensive dialysis or a Hi-Trap G25 column, equilibrated with 0.15 M NaCl saline. When desalting using G25 column, care was taken to avoid collecting fractions with residual ADH. The level of hydrazide incorporated to the polysaccharide was determined by TNBS assay to give the extent of polysaccharide activation.

As another proxy for the level of dextran activation, we used the extent of conjugation of TNP_1_-BSA to Dextran T2000, monitored by SEC HPLC. SEC HPLC was performed on a Waters Alliance System (Milford, MA, USA) system connected with Tosoh Biosciences (King of Prussia, PA, USA) G4000 SEC column (7.8 × 300 mm) equilibrated with phosphate buffered saline plus 0.02% sodium azide at a flow rate of 1 mL/min. Detection of TNP-BSA was at 420 m. Since T2000 dextran is of an average MW of 2000 kDa; its conjugate with TNP-BSA was well separated from the unconjugated protein (MW 66 kDa). The percent of high molecular weight protein (protein coeluting with or ahead of the starting dextran) was used to estimate the extent of conjugation. This assumption was confirmed by analyzing several conjugates fractionated on a preparative Sephacryl S-400HR^®®^ SEC column (Cytiva, Marlborough, MA, USA) (data not shown).

## 3. Results

### 3.1. CDAP Stability in Water and Buffered Solutions

We first undertook a study of CDAP stability in aqueous solutions. We previously observed that an unbuffered aqueous solution became acidic on the addition of CDAP [10], with the rate of acidification dependent on the CDAP concentration (Appendix A). Following an initial rapid decrease in pH, the rate of pH change slowed significantly. This observation is consistent with the reported acid stability of CDAP [18] and explains how CDAP essentially “self-stabilizes”. As the solution is unbuffered initially, only a negligible amount of CDAP needs to hydrolyze to reduce the pH. In the 1996 protocol [10], CDAP was added to the aqueous polysaccharide solution and the pH was raised 30 s later. However, we have found that the aqueous CDAP solution was reasonably stable as the extent of protein conjugation to CDAP-activated dextran was comparable, whether the pH was raised at 30 s or 5 min after CDAP addition (data not shown). Therefore, the exact timing for pH adjustment is not that critical but should be consistent. The main purpose of waiting after adding the CDAP is to allow the solution to be mixed prior to beginning the activation.

### 3.2. Monitoring CDAP by UV Absorbance

We used an absorbance method to measure the concentration of CDAP in a solution. We observed that the UV spectrum of CDAP in 0.1M HCl was unchanged after 16 h of incubation at room temperature, indicating that the reagent was stable in 0.1 M HCl, consistent with Kohn and Wilchek’s finding of a half-life of “several weeks” in 0.1M HCl [18]. CDAP and DMAP (dimethylaminopyridine), its hydrolysis product, have distinct UV spectra, with peaks at 301 and 280 nm, respectively, in 0.1 M HCl [18]. In base, DMAP is not protonated and the peak shifts to 260 nm. The UV spectrum of CDAP in 0.1 M HCl is provided in Appendix A, along with those of DMAP in both acid and base. DMAP has only about 5% of the absorbance of CDAP at 310 nm. Over the course of this work, we used the decrease in absorbance in the region 310–320 nm, wavelengths with minimal DMAP absorbance, to monitoring CDAP hydrolysis.

### 3.3. CDAP Stability as a Function of pH

To determine the stability of CDAP as a function of pH, CDAP was added to buffered solutions of distinct pH values, aliquots periodically transferred to 0.1 M HCl and the absorbance at 312 nm measured and the % CDAP remaining calculated (Figure 3a). CDAP was stable in 0.1 M HCl and quickly hydrolyzed after exposure to 0.1 M NaOH. We observed that CDAP hydrolysis sharply increased around pH 8. At pH 9, the standard pH for activation, greater than 95% of the CDAP had hydrolyzed within 3 min. We confirmed these results by directly assaying for CDAP using the dimethylbarbituric acid assay [19] (data not shown). In a separate experiment, we assayed for CDAP at 310 nm after 30 s at each pH (Figure 3b). At pH 8, only about 10% of the CDAP hydrolyzed while at pH 9 about 90% had hydrolyzed within 30 s. The rapid hydrolysis observed at pH 9 means that most of the CDAP is consumed by hydrolysis, requiring a large excess of reagent in order to obtain sufficient polysaccharide activation.

The rapid hydrolysis of CDAP in alkaline buffers, especially above pH 8, means that even small deviations of pH will have a marked effect on CDAP hydrolysis, and thus the amount of CDAP available for activating the polysaccharide. Therefore, for reliable polysaccharide activation, careful control of pH is necessary to prevent overshooting the target pH. Variations in how the pH is raised, the rate of increasing the pH, whether there are pH “hotspots” in the solution and whether the pH overshoots the target could all affect CDAP stability and, therefore, the reproducibility of polysaccharide activation. It should be noted that the rate of reaction of CDAP with the polysaccharide, along with the stability of the activated polysaccharide, is also pH-dependent, so pH excursions result in inconsistent activation. We have observed that the polysaccharide solution may become opalescent or even gel-like, if pH spikes occur, presumably due to the reaction of adjacent hydroxyls with activated sites.

### 3.4. CDAP Stability as a Function of Temperature

Having identified that CDAP rapidly hydrolyzes at pH 9 at room temperature, the standard activation conditions, we next investigated the hypothesis of slowing down CDAP hydrolysis by reducing the temperature. Neither of our earlier publications on CDAP chemistry emphasized performing the activation in the cold and all but one of the experiments described (e.g., Lees [10], Figure 3) were at room temperature. To perform the temperature study experiment, CDAP was added to an unbuffered aqueous solution at room temperature or prechilled in an ice water bath, and the temperature was kept constant. At 30 s, the pH was raised to 9 with 0.1 M NaOH. Aliquots were periodically transferred to 0.1 M HCl and the amount of residual CDAP determined from the absorbance at 310 nm. Comparing with room temperature, CDAP hydrolysis was found to be significantly slower in the cold (Figure 4). The figure shows that at pH 9, virtually all the CDAP was consumed within about 1 min at room temperature but a significant amount of CDAP still remained even after 6 min at 0 °C.

We evaluated CDAP stability at an even lower temperature by chilling a saline solution of dextran to −10 °C while performing the activation. CDAP hydrolysis was indeed slowed down even further (data not shown). We did not further explore these conditions, however, as the inconvenience seemed to outweigh possible advantages.

### 3.5. Reaction of CDAP with Polysaccharides at Lower Temperature

Reducing the rate of CDAP hydrolysis by lowering the reaction temperature increased the amount of reagent available for activation. However, it was a concern that lower temperatures would also slow the reaction of CDAP with the polysaccharide. We indirectly estimated the rate of polysaccharide activation by monitoring the consumption of CDAP in the presence and absence of dextran. We found that the absorbance at 310 nm decreased faster in the presence of polysaccharide than without, indicating more rapid consumption of CDAP in the presence of dextran at both room temperature and 0 °C. The increased consumption of CDAP in the presence of dextran was barely detectable at room temperature where hydrolysis was already rapid (not shown). At 0 °C, an enhanced rate of CDAP consumption was observed in the presence of dextran (Appendix A). Increasing the polysaccharide concentration increased the rate of CDAP consumption. Thus, the activation rate did not appear to be unduly sluggish when performing the reaction at 0 °C.

In addition to CDAP being more stable in the cold, we have also found that the activated polysaccharide was more stable at 0 °C than at room temperature (Lees, unpublished, manuscript in preparation), allowing higher levels of conjugation when the process was carried out in the cold. As an example, we found that using an activation protocol with 0.4 mg CDAP per mg dextran, 45% and 66% of BSA was conjugated with room temperature and 0 °C activation, respectively; it took double that amount of CDAP (0.8 mg/mg dextran) to achieve a similar level of conjugation (70%) using room temperature activation (data not shown).

### 3.6. Functionalization of CDAP-Activated Dextran with Diamines

Polysaccharides derivatized with amines provide a convenient route to conjugation, as they are easily functionalized using NHS ester reagents [20]. Hydrazide-derivatized polysaccharides can also be used in carbodiimide conjugation. In our earlier work, we were unsuccessful in derivatizing CDAP-activated dextran with ethylenediamine and proposed that it was due to the reaction of the second amine with the isourea group on the transiently-aminated polysaccharide [16]. To further explore this point, we reacted CDAP-activated dextran with aliphatic terminal diamines of ethane, propane, butane and hexane, as well as adipic dihydrazide. Following desalting, the amine or hydrazide to dextran ratio was determined. We found that we could detect few primary amines on the polymers when ethylene diamine or propane diamine was used. In contrast, diamines of butane and hexane, as well as adipic dihydrazide, yielded dextrans with high levels of amines or hydrazides (Appendix A). We previously found that hydrazine could be successfully used to derivatize CDAP-activated dextran [16]. As only ethylene diamine and propane diamine, but not the other diamines, can form energetically favorable five- and six-member rings with the isourea bond, it seems likely that this is a reasonable explanation for the lack of amination with these two diamines.

### 3.7. Effect of Buffer Salts on CDAP Stability

It can be challenging to both reproducibly adjust the pH of an unbuffered CDAP solution and reproducibly maintain the pH with only NaOH, as CDAP hydrolysis releases acid and there is no buffer to neutralize it. Given that pH control is critical to reproducible activation, an obvious solution is to prepare the polysaccharide in a buffered solution. This protocol modification would provide a definitive start time to the activation reaction and facilitate pH maintenance. However, in previous work at room temperature, we found that activation in the presence of buffers often failed or was much lower than expected. Polysaccharide solutions prepared in water, saline (0.15 M NaCl) and 2 M NaCl have been successfully activated with CDAP and conjugated to proteins [10,16,21]. Consistent with this observation, we determined that the rate of CDAP hydrolysis was unchanged in the presence and absence of 1 M NaCl (not shown). To evaluate reagent stability in the presence of buffers we measured residual CDAP after 10 min in 0.1 M HEPES, pH 8.3 solution at 0 °C, alone or with 25 mM borate, sulfate, phosphate or bis-Tris (Figure 5). We found that the presence of phosphate ion rapidly hydrolyzed CDAP. The other salt components also accelerated CDAP hydrolysis at pH 8.3, 0 °C relative to HEPES, but to a much lesser extent than phosphate. Clearly, phosphate should be avoided for CDAP activation reactions.

As we had used borate buffer in our pH stability study (Figure 3), it was possible that borate ion, not pH was responsible for accelerated CDAP hydrolysis at pH 9. To directly evaluate whether borate was detrimental, we compared CDAP activation of dextran at pH 9 and 0 °C with either 0.1 M borate or DMAP/NaOH as the base. At 10 min, the CDAP-activated dextran was reacted with ADH and desalted, and the hydrazide:dextran ratio was determined. The level of derivatization decreased by 28% when borate buffer was used. Triethylamine (TEA), the base originally used for CDAP activation, has a pKa of 10.75, and thus provides poor buffering activity at pH 9, particularly since the reaction generates acid. Bicine, with a pKa of 8.35, seemed promising, as it is a tris-amine and should have reduced nucleophilicity. However, we found that activation in a 0.1 M bicine buffer at pH 9 and 0 °C reduced the level of ADH derivatization by 45% compared with the use of DMAP/NaOH as the base (see next section). Thus, while buffers like bicine and borate can be used to fix the reaction pH, more reagent will be required to achieve the same level of activation. We note here that phosphate did not impact the stability of CDAP-activated dextran, unlike its significant acceleration of CDAP hydrolysis (Lees, unpublished data).

### 3.8. Use of DMAP to Manage pH

We observed that while performing activation at decreasing CDAP concentrations, increasing care was needed to avoid overshooting the target pH. We suspected that the DMAP released by partial CDAP hydrolysis acts as a buffer. With the slower hydrolysis at 0 °C, along with less CDAP being required for activation, the resulting lower DMAP concentration provided reduced buffering power. It occurred to us that additional DMAP added to the polysaccharide solution could improve pH control because DMAP has a pKa of 9.5 in aqueous solutions. We found that adding DMAP to the polysaccharide solution indeed provided additional buffering power and facilitated pH control. Furthermore, instead of NaOH, a preadjusted DMAP stock solution can be used to raise the pH, allowing the target to be precisely achieved [15]. In our lab, even an inexperienced scientist could achieve consistent activation results, as the DMAP buffering removed the challenge of targeting and maintaining the pH. A DMAP buffer was particularly useful for development work (e.g., design of experiment studies) which can involve small volumes and a large number of samples.

The preparation of the concentrated DMAP stock solution was empirical, as the pH of a DMAP solution increases significantly upon dilution and is dependent on the ionic strength and temperature. We first prepared a 2.5 M DMAP solution of about pH 8.5, and then diluted, at the required dilution ratio, into a solution at approximately the same temperature and ionic strength as that of the polysaccharide solution. The pH of the DMAP stock was iteratively adjusted until the diluted DMAP solution matched the target activation pH. While rather clumsy, this preparation method of the DMAP stock solution allowed for the reaction pH to be accurately set with little or no further addition of base needed to maintain the pH. We found 50–250 mM DMAP provided sufficient buffering power, depending on the amount of CDAP used.

We evaluated if there was an advantage of using DMAP to preadjust the polysaccharide solution pH prior to the addition of CDAP. For this comparison, dextran was activated with 0.5 mg CDAP/mg polysaccharide at 0 °C in the presence of DMAP buffer. DMAP was added prior to or after the addition of 0.5 mg CDAP/mg dextran. In each case, the final DMAP concentration was 50 mM and the pH maintained at 9 with 0.1 M NaOH. After 15 min, an equal volume of 20 mg/mL TNP-BSA was added to the activated dextran solution, and conjugation was allowed to proceed overnight. Conjugates were analyzed through SEC HPLC analysis. We found identical levels of high molecular weight conjugate (66.5%) for the two methods, as shown in Figure 6, indicating that preadjusting the polysaccharide solution pH with DMAP, prior to the addition of CDAP, was an effective method to control the activation pH and to achieve efficient conjugation.

Preadjusting the pH using the DMAP stock eliminated much of the variability inherent in the original CDAP protocol and minimized the risk of overshooting the target pH, as both adjusting and maintaining the pH became much easier and more effective. A further advantage of using DMAP as the buffer is that it does not introduce any new chemical entities into the reaction mixture as the chemical is already a process residual.

### 3.9. Stability of CDAP-Activated Dextran as a Function of pH

As outlined in Figure 2, in addition to CDAP hydrolysis and reactivity, the stability of the activated polysaccharide is also important for achieving good conjugation. Having shown that at pH 9 and 0 °C, CDAP stability increased without reactivity being impaired, we next evaluated the stability of the activated polysaccharide as a function of pH. It has been reported that the cyanoester on CNBr-activated Sepharose^®®^ resin was most stable in dilute acid [22]. To explore the pH stability of CDAP-activated dextran, we incubated activated dextran at a range of pH for 2 h on ice, and then derivatized an aliquot with ADH. Following desalting, the hydrazide to dextran ratio was determined. Unlike CNBr activation, we did not observe a pH range that promoted the stability of the activated carbohydrate, as we found that, at all pH values tested, the extent of derivatization decreased by about the same amount, 40–65%, relative to activated dextran that was immediately reacted (Appendix A).

### 3.10. CDAP Stability as Function of pH

We next examined CDAP stability as a function of pH at 0 °C. CDAP at 2.5 mg/mL was incubated in ice-cold buffers over pH 5–9 and aliquots periodically transferred to 0.1 M HCl to prevent further hydrolysis of the reagent. Absorbance at 320 nm was used to monitor CDAP hydrolysis (Figure 7a). As we observed at room temperature, there was also a rapid increase in the rate of CDAP hydrolysis between pH 8 and 9. It took ~10 min for CDAP to be consumed at pH 9, >60 min at pH 8 and >4 h at pH 5–7 (not shown). The % CDAP remaining at 15 min is replotted in Figure 6b. Comparing Figure 3b and Figure 7b, the pH dependence of CDAP hydrolysis at 0 °C is similar to that at room temperature, but the rate is 10–15 times slower in the cold. As noted earlier, the use of borate buffer at pH 9 accelerated CDAP hydrolysis.

### 3.11. Dextran Activation Time as a Function of pH

We previously reported that at pH 9 and room temperature, the optimum activation time was 2.5 min [10]. Our observation of increased stability of CDAP at low temperature suggested that at 0 °C, the longer reagent lifetime, and possibly longer lifetime of the activated polysaccharide, would change the optimal pH and activation time. However, there was the possibility that the reactivity of polysaccharide hydroxyls would be reduced at a less basic pH. To determine the relationship between optimum activation time and pH, dextran was activated at pH 7–9.5 at 0 °C using DMAP/NaOH as the base. Aliquots of the activated dextran were periodically added to an equal volume of 0.5 M ADH, incubated for 2 h and desalted, and the hydrazide:dextran ratio was determined. The results are presented in Figure 8, panel A. At pH 9, the optimum activation time increased from ~2.5 min at room temperature to 10–15 min at 0 °C. With activation at pH 8, maximum derivatization occurred at about 1 h and the level of derivatization declined thereafter, but less rapidly than at pH 9. At pH 7, the level of derivatization was still increasing after 2 h of activation. At pH 9.5, increased reactivity with hydroxyls was counterbalanced by more rapid CDAP hydrolysis, resulting in a shorter optimum activation time and lower levels of derivatization. Figure 8, panel B shows the peak level of derivatization as a function of time, along with the activation time to reach that level. At 0 °C, the optimal pH for dextran remained at 9. Thus, good levels of derivatization were found at pH 7–9.5, but the optimal activation time was highly dependent on the pH, reflecting its effect on the balance of CDAP’s half-life and reactivity.

We also used the indirect method described in Figure 6 to assess the level of dextran activation. This method, in which the dextran was activated under various conditions and then conjugated to protein under a standard condition, is a convenient proxy for the level of activation. For this study, dextran buffered at pH 5–9 on ice, was activated with 1 mg CDAP per mg dextran and the pH maintained with dilute NaOH. At various times, an aliquot of the activated dextran was withdrawn and added to a solution of TNP_1_-BSA buffered at pH 9. Conjugation was allowed to proceed overnight to minimize the differences in conjugation time. A control was included by transferring the same volume of aliquot from the time zero activation to the protein, to determine the amount of conjugation due to transfer of residual CDAP to the pH 9 conjugation buffer. We found about 15% high MW conjugate, with the time zero transfer. The results are presented in Figure 8, panel C. Little conjugation was found when activation was performed at pH 5 or 6, even with up to 3 h of activation. At pH 7, the level of conjugation was still increasing at 3 h. We did not evaluate longer activation times. At pH 8, peak conjugation occurred at 60 min while at pH 9 maximum conjugation was with 15 min of activation. We observed a higher level of conjugation at pH 8 than at pH 9 due to the accelerated CDAP hydrolysis by the borate buffer, as discussed earlier.

To determine if conjugation with pH 8 activation could be further increased, dextran was activated at this pH using 0.5, 1 or 1.5 mg CDAP/mg dextran at 0 °C. At 1 h an aliquot was reacted with TNP_1_-BSA and the extent of conjugation analyzed by SEC HPLC. We found that the extent of conjugation was 41, 60 and 76%, respectively. For comparison, dextran activated at pH 9 with 0.5 mg/mg CDAP had 55% TNP_1_-BSA conjugated. Thus, high levels of conjugation could be achieved at pH 8 by increasing the amount of CDAP, allowing pH sensitive polysaccharides to activated to high levels.

### 3.12. Activation of pH Sensitive Polysaccharides

The activation and conjugation pH used in the original CDAP activation and conjugation protocol (pH 9–10), may still cause hydrolysis of some polysaccharides. *Neisseria meningitidis* serotype A (MenA), Hib PRP and pneumococcal polysaccharide serotype Pn19A are polysaccharides with a pH-sensitive phosphodiester linkage and are known to depolymerize at pH 9, but are also known to be significantly more stable at lower pH [23]. To evaluate the activation of these polysaccharides, MenA was activated with CDAP at pH 8 and 0 °C and aliquots were periodically transferred to an equal volume of 0.5 M ADH at pH 8. Following a 1 h reaction, samples were desalted and the hydrazide to MenA ratio was determined and correlated with the activation time course (Figure 9). As with dextran, approximately 1 h was the optimum activation time under these reaction conditions, achieving about 20 hydrazides per 100 kDa of polysaccharide. In a separate set of experiments, we activated Pn19A and Hib PRP with CDAP at pH 8 for 1 h at 0 °C and then derivatized the polysaccharides with ADH at pH 5. After desalting, we found 15 and 24 hydrazides per 100 kDa of Pn19A and Hib PRP, respectively. These ratios are comparable to those reported in the literature for other ADH-polysaccharides [13,24,25]. Thus, pH-sensitive polysaccharides can easily be functionalized at pH 8 using modest CDAP to polysaccharide ratios (0.5 mg/mg).

## 4. Discussion

The use of CDAP in the synthesis of conjugate vaccines has been found in the literature for more than 25 years [10,16] and the chemistry is used in the preparation of licensed pneumococcal and meningococcal vaccines as well as a number of vaccines in clinical development. As the best practice of its use has not been well described in the literature, we undertook the study to optimize the activation of polysaccharides with CDAP. A separate study will address methods to optimize conjugation to CDAP-activated polysaccharides (Lees, manuscript in preparation). Dextran and BSA were used as a model system because these components are low cost and have an abundance of hydroxyls for activation and primary amines for conjugation, respectively, making them perhaps the easiest pair to conjugate using CDAP chemistry. While the specifics may not apply to other polysaccharides, our work is intended to provide a guide to CDAP conjugation with more clinically relevant carbohydrates and carrier proteins.

In our original protocol [10], polysaccharide activation with CDAP was initiated by raising the pH with a single addition of TEA as the base, generally to a target pH in the range of 9–10. The importance of controlling the pH was not emphasized, although other publications noted that the pH had been maintained [21]. The activation in the original protocol was rapid, ~2.5 min, resulting in a process that could be frenetic and difficult to control. It was likely that the pH exceeded the target (Figure 10, solid line). It also made scale-up challenging since the reaction time remained short while the volume of liquids to be added and mixed increased with scale. In this work, we show that at pH 9 at room temperature, not only is the polysaccharide activation extremely rapid, but the rate of CDAP hydrolysis is extremely rapid as well. This makes precise control of pH absolutely critical to uniform activation of the polysaccharide. Since CDAP hydrolysis also releases acid into the solution, in the absence of added buffer, it can be difficult to raise and maintain the activation reaction at the target pH value. However, we found that not all buffers are suitable to be used in the activation reaction. Some buffers promote CDAP hydrolysis and activation levels are reduced in their presence. Phosphate ion was found to rapidly hydrolyze CDAP and should be avoided. Sulfate, borate, bis-Tris and bicine were all found to reduce levels of activation and if they are used, more CDAP will be needed to compensate for the increased hydrolysis. The use of dilute buffers (e.g., 0.1–0.25 M NaOH) to increase the pH in a controlled manner resulted in more consistent activation, as illustrated in Figure 10 (dashed line).

CDAP will react with thiols [26], glycine and other nucleophiles found on proteins (Lees, unpublished results). To avoid protein modifications that could introduce unwanted epitopes, it is important that all unreacted CDAP be consumed by hydrolysis or reaction prior to the addition of the protein. Under the original activation protocol, (pH 9, room temperature and 2.5 min activation) all of the CDAP should be consumed prior to the addition of protein. However, when performing direct conjugation of proteins to CDAP-activated polysaccharide at lower activation pH and temperature, care should be taken there is no active reagent at the time the protein is added.

We found that pH control became more difficult when using less CDAP and concluded that DMAP, the hydrolyzed product from CDAP, was acting as a buffer. At low temperature, with less CDAP needed and slower hydrolysis, there was an insufficient quantity of DMAP present. We compensated for this by adding additional DMAP base to the polysaccharide solution. The pKa of aqueous DMAP is about 9.5 [27] and the added DMAP contributes additional buffering capacity to assist in pH control. Since the activation pH is below the pKa, DMAP buffer helps to reduce the tendency to overshoot the target pH. In the published protocol, CDAP is added to the solution first and the pH then raised. We found that DMAP, unlike many other buffers, which negatively affect the activation level, can be used to pre-adjust the pH of the polysaccharide solution prior to the target activation pH prior to adding CDAP, without impacting the level of activation. This protocol only requires the maintenance of the activation pH (Figure 10, dotted line) versus the original protocol which involved both raising and maintaining the pH. An additional advantage for DMAP over other bases is that as it is already a reaction product, no further analytical techniques are needed to detect process residue. This method has been successfully used to conjugate *Salmonella typhimurium* polysaccharide to flagellin [15].

CDAP hydrolysis is markedly slower in the cold, increasing the amount of active CDAP available for polysaccharide activation. We found that less CDAP was needed at 0 °C to achieve the same level of activation compared with the same reaction at room temperature. With the longer half-life of CDAP in the cold, the optimal activation time at pH 9 increased from ~2.5 min to 10–15 min. This provided more time for the pH to be raised and made it much easier to avoid overshooting the target. The marked pH dependence of CDAP hydrolysis makes control of this parameter critical to achieving reproducible activation. Good mixing while adjusting the pH is necessary to avoid “hotspots” and it is helpful if the pH probe and meter have a rapid response time. We suggest that it is better to slowly and consistently raise the pH to the target value rather than attempt to do it rapidly, which risks overshooting the target pH. Performing the reaction at 0 °C slows down acid production, facilitating pH adjustment. The longer activation time provides a wider window for pH adjustment and makes the process less frenetic, more reproducible and easier to scale-up.

We previously reported that the optimal pH for CDAP activation of dextran at room temperature was around pH 9 when the activation time was fixed at 2.5 min [10]. Considering the strong pH and temperature dependence of the reagent’s half-life, the optimal pH and time would change when the activation time and/or then activation temperature was varied. For example, when the activation was in the cold, the optimal activation time at pH 9 increased to 10–15 min (see above). By prolonging the activation time to 3 h or longer, we found that CDAP could efficiently activate dextran even at pH 7. At pH 8 at 0 °C, the optimal activation time was around 1 h. In case a condition of interest does not lead to the desired level of activation, one has the option to increase the amount of CDAP used. For example, when carrying out activation at 0 °C, by increasing the amount of CDAP, we could achieve a comparable level of conjugation at pH 8 to that obtained from a pH 9 activation.

Another advantage of activation at below pH 9 is the likely reduction in the carbamate and imidocarbonate side reactions described by Kohn and Wilchek [18], as these are promoted by high pH. These side reactions reduce the efficiency of activation and introduce new epitopes into the polysaccharide. Furthermore, cyclic imidocarbonates, which may crosslink polysaccharide polymers, can lead to viscous solutions that are difficult to filter. Activation below pH 9 also enables a strategy for conjugating base-sensitive polysaccharides. In this work we have shown that high levels of polysaccharide activation can be achieved at pH 8 at 0 °C, if the optimal time for activation is used. Our previous work found that the hydrazides (either as adipic dihydrazide or as hydrazide derivatized protein) will react with CDAP-activated polysaccharide essentially independent of pH [16]. Thus, base-sensitive polysaccharides, such as meningitides serotype A, Hib PRP, pneumococcal serotypes Pn6A and 19A can be derivatized using CDAP chemistry at pH 8.

Kohn and Wilchek determined that a weakly reactive imidocarbonate may be a significant intermediate with CNBr activation, due to the high pH required [9] but its role in CDAP activation is uncertain. These authors suggested that the active intermediate of CDAP-activated Sephadex may be a pyridinium-isourea [18], (Figure 11) rather than a cyanate ester.

We have observed that CDAP-activated dextran, when desalted on a G-25 column, exhibited a large absorbance peak at 280 nm co-eluting in the dextran fraction, whereas only a small absorbance peak was observed if the activated dextran had been first reacted with ADH. When this desalted CDAP-activated dextran was treated with ADH, the 280 absorbing material was found to be permeable to a 3K membrane. This supports the finding of Kohn and Wilchek that the pyridinium-isourea contributes to the activation of the polysaccharide. However, the relative role of this intermediate versus the contribution from cyano-esters is uncertain. The presence of the pyidinium-isourea intermediate demonstrates the need for proper quenching of the reaction product to ensure its displacement from the polysaccharide following conjugation.

### A Practical Approach to CDAP Activation of Polysaccharides

A consistent activation protocol is necessary for the reproducible synthesis of conjugates using CDAP chemistry. Precise control of pH is critical, which is facilitated by the use lower temperature and pH. It is our experience that it is better to remain below the target pH 9, than to exceed it as the high pH promotes hydrolysis. We prefer to smoothly raise the pH over 1–2 min than all at once. In our laboratory we use an electronic titrator or pipettor to add the base in a controlled manner. In the development of an activation protocol, we begin with producing data at a target pH of 8 and 9 as shown in Figure 11. The reactions are performed on ice, with preadjustment of the polysaccharide to the target pH using DMAP buffer, so that only pH maintenance necessary after adding CDAP. Aliquots from the reaction are periodically transferred to solutions of ADH and the hydrazide/polysaccharide ratio of the derivatized polysaccharides are determined after desalting. The time course for activation is also followed by the proxy method of protein conjugation monitored using SEC HPLC. This combination allows for the rapid determination of the necessary level of activation to achieve good conjugation. In this work, we only evaluated pH 8 and 9 but process optimization using design of experiment (DOE) can allow fine tuning of the activation parameters. Polysaccharide activation is the first phase of CDAP chemistry. Optimizing the conjugation step will be addressed in a separate publication.

## 5. Conclusions

CDAP chemistry provides a simple method to activate polysaccharides for functionalization or direct conjugation of proteins. Key points for CDAP activation of polysaccharides are summarized: in Box 1.

Box 1Key points for optimizing CDAP activation of polysaccharides.

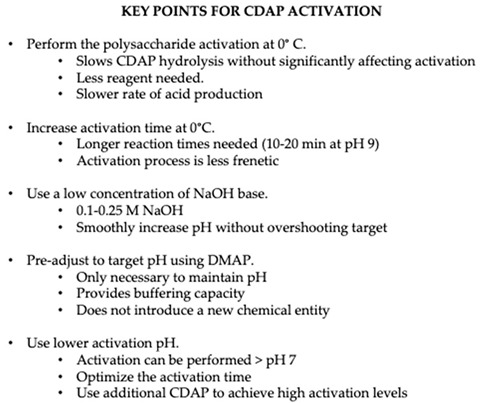



Control of pH is key to reproducible CDAP polysaccharide activation and we found that performing the reaction in the cold slows CDAP hydrolysis, leaving more reagent available for reaction, while having a minimal effect on the rate of activation. DMAP was found to be an ideal buffer to use to facilitate pH control. Caution needs to be exercised when using other buffers to avoid potential negative effect on the activation. Performing cold activation slows CDAP hydrolysis, leaving more active reagent available for activation, with minimal effect found on the level of activation when longer activation times were used. Good levels of polysaccharide derivatization could be obtained even at pH 7 with longer activation times. The longer reaction times allow achieving same high levels of activation with less CDAP and make the process easier to control and practical to scale-up.

## Figures and Tables

**Figure 1 vaccines-08-00777-f001:**
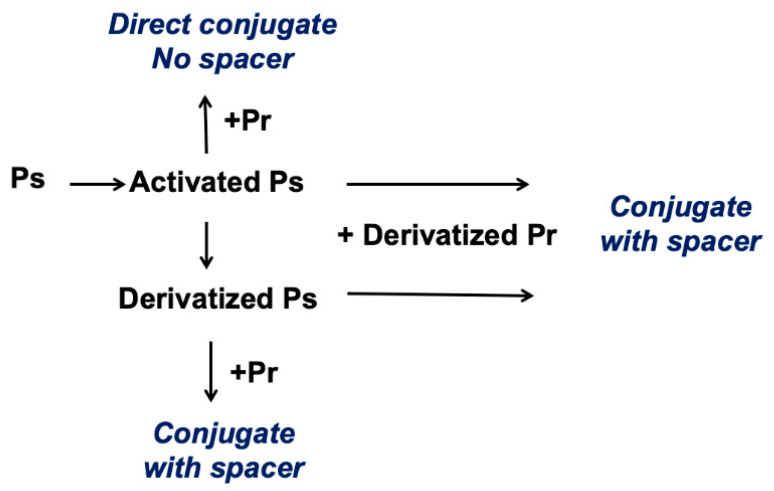
Overview of conjugation of protein to polysaccharide.

**Figure 2 vaccines-08-00777-f002:**
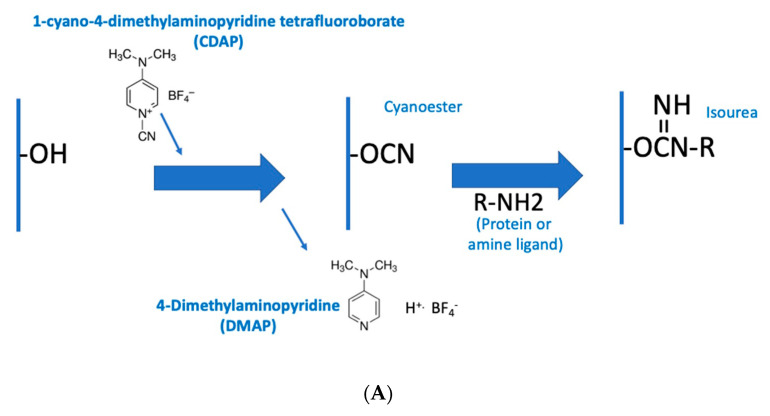
Process of CDAP activation and conjugation, conceptually dividing the two phases. Overview (**A**) and details (**B**) of CDAP activation and conjugation with the chemical structures of CDAP and DMAP, the hydrolysis product.

**Figure 3 vaccines-08-00777-f003:**
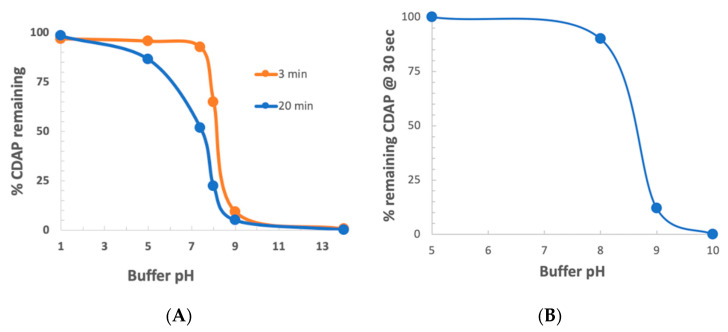
% CDAP remaining as a function of pH. (**A**) % CDAP remaining at 3 and 20 min as a function of pH at room temperature, assayed by absorbance at 312 nm. (**B**) % CDAP remaining at 30 s at room temperature, assayed by absorbance at 310 nm. 100% = CDAP transferred to 0.1 M HCl only. All buffers are 0.1 M: pH 1 (HCl), pH 5 (sodium acetate), pH 7 and 8 (sodium HEPES), pH 9 (sodium borate) and pH 10 (sodium carbonate).

**Figure 4 vaccines-08-00777-f004:**
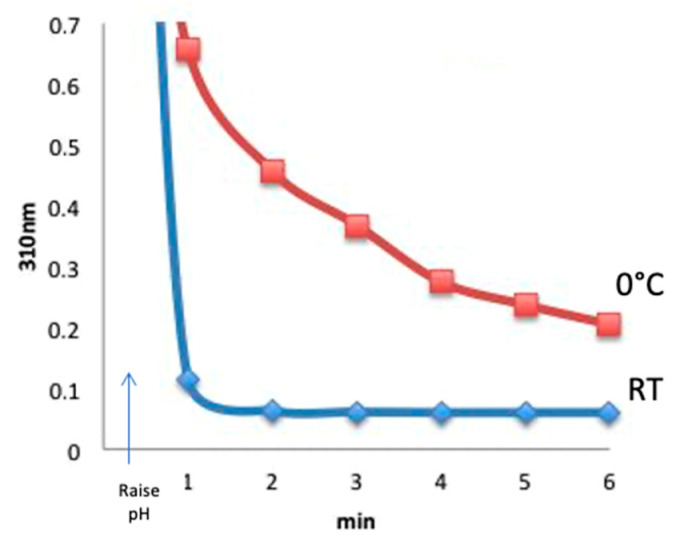
CDAP remaining at room temperature vs. 0 °C. CDAP was added to an aqueous solution at the indicated temperature. At 30 s, the pH was raised to 9 with 0.1 M NaOH. Aliquots were transferred to 0.1 M HCl and the absorbance read at 310 nm.

**Figure 5 vaccines-08-00777-f005:**
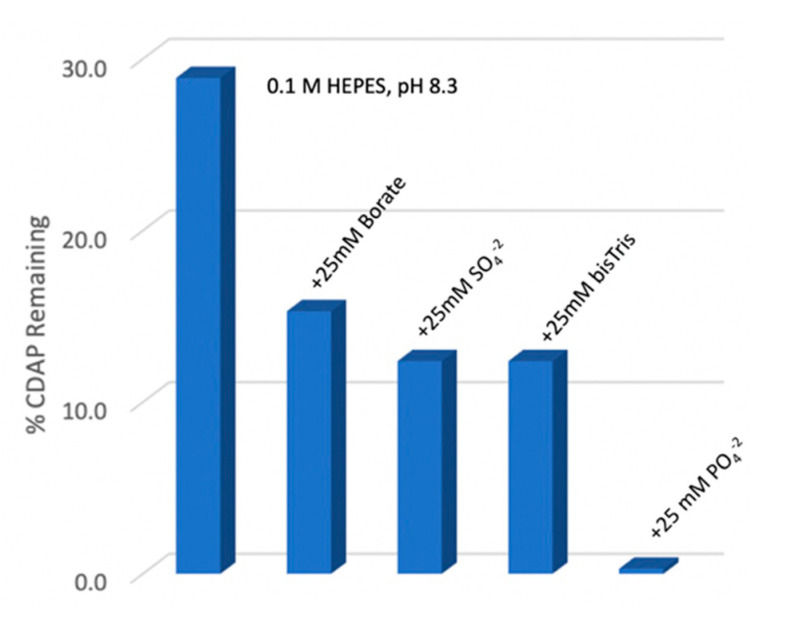
CDAP stability in buffers and salts. 0.1 M HEPES solutions (1 mL) containing 25 mM sodium phosphate, sodium sulfate, sodium borate, bis-Tris or no added salt, each with a final pH of 8.3 at 0 °C were prepared. 25 µL 100 mg/mL CDAP was added, and at 10 min, a 25 µL aliquot was transferred to 2 mL 0.1 M HCl. Controls for 0% and 100% hydrolysis were obtained by adding CDAP to 0.1 M HCl or 0.1 M NaOH and each were subsequently transferred to 0.1 M HCl. All samples were read at 312 nm and the % remaining CDAP was calculated.

**Figure 6 vaccines-08-00777-f006:**
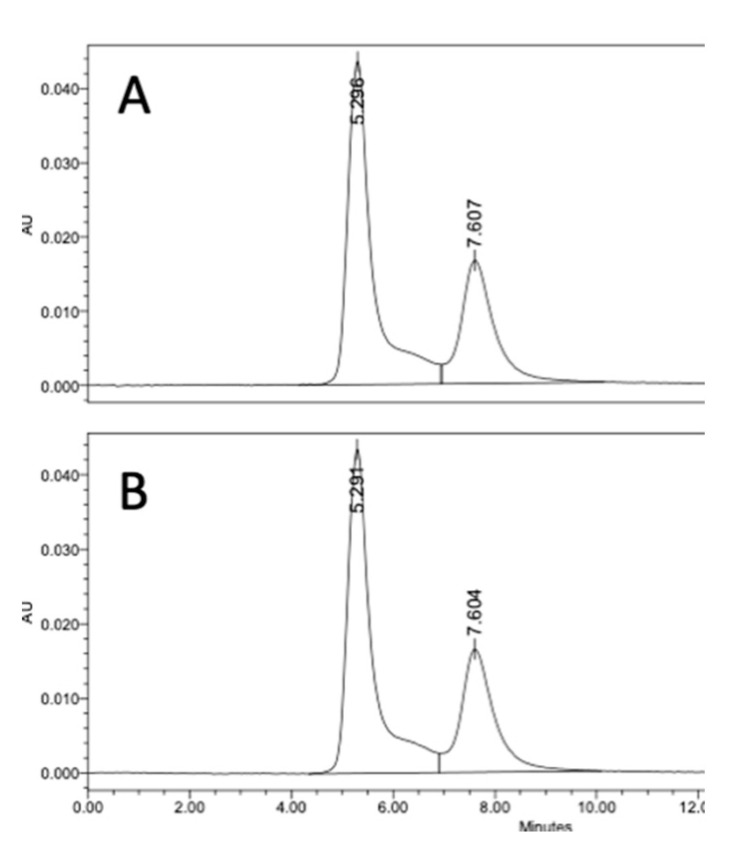
Activation of dextran in DMAP buffer. DMAP was added prior to (**A**) or after (**B**) the addition of 0.5 mg CDAP/mg dextran. The pH was maintained at pH 9 with 0.1 M NaOH for 15 min followed by conjugation to TNP-BSA. The reaction was maintained on ice. Conjugates were analyzed using a Tosoh (King of Prussia, PA, USA) G4000 SEC HPLC column, with detection at 420 nm. The high molecular weight protein was considered conjugate.

**Figure 7 vaccines-08-00777-f007:**
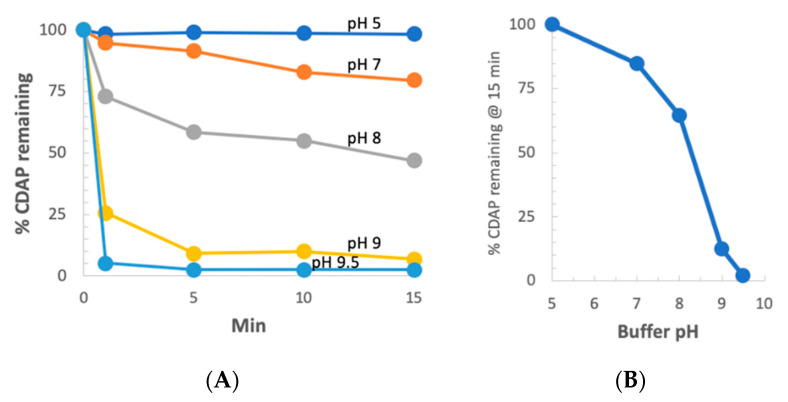
CDAP stability at 0 °C vs. pH. (**A**) CDAP was added to ice-cold solutions of 0.1 M buffer at pH 5–9 to a concentration of 2.5 mg/mL. At various times, a 25 µL aliquot was withdrawn and added to 975 µL of 0.1 M HCl and the absorbance read at 320 nm. (**B**) plot showing the data from Panel A at 15 min as a function of pH. The following 0.1 M buffers were used: HCl (pH 1), sodium acetate (pH 5), MES (pH 6), HEPES (pH 7 and 8) and sodium borate (pH 9 and 9.5).

**Figure 8 vaccines-08-00777-f008:**
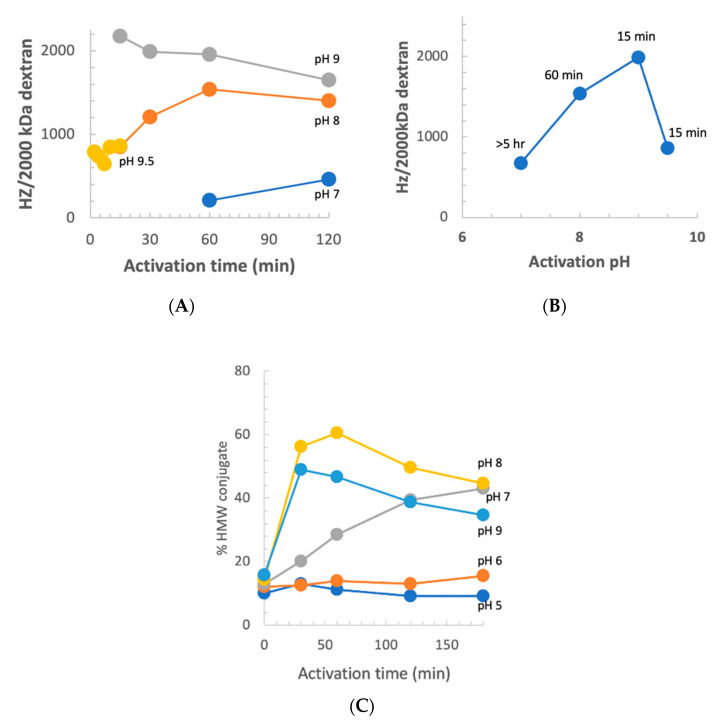
CDAP activation of dextran at pH 5–9.5 as a function of time at 0 °C. (**A**) ADH-derivatized dextran. ADH was reacted with 2000 kDa dextran activated at pH 7–9.5 using 0.5 mg CDAP/mg dextran with DMAP and NaOH to maintain the reaction pH. (**B**) Peak level of derivatization vs. activation pH. Peak activation time is indicated above each point. (**C**) TNP-BSA-conjugated dextran. Dextran was activated using 0.5 mg CDAP/mg dextran at pH 5–9 using buffers to maintain the pH. The activated dextran was reacted with TNP_1_-BSA overnight at pH 9 and then analyzed by SEC HPLC. The level of dextran activation was determined from the % of high molecular weight protein. 0.1 M buffers: pH 5 sodium acetate, pH 6 MES, pH 7 and 8 HEPES and pH 9 sodium borate. At time 0, about 15% of the BSA was conjugated.

**Figure 9 vaccines-08-00777-f009:**
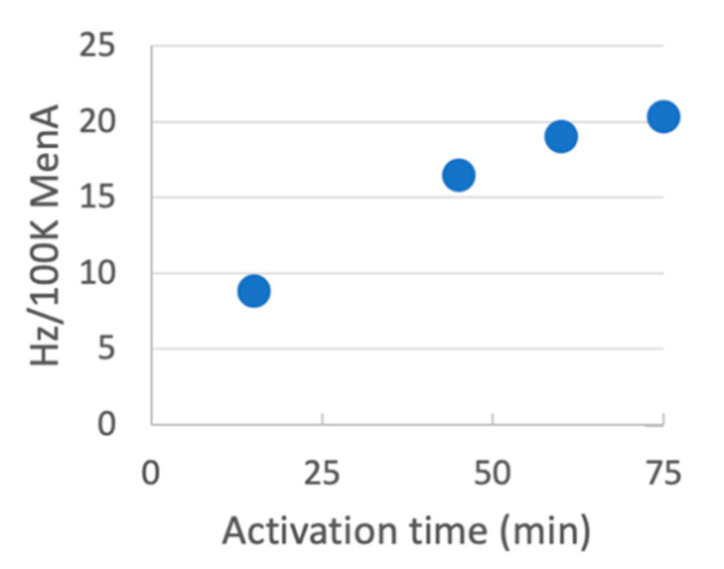
Time course of activation of MenA polysaccharide at pH 8 at 0 °C. 5 mg/mL MenA was brought to pH 8 with DMAP at 0 °C and CDAP added to 2.5 mg/mL. The pH was maintained at 8 with 0.1 M NaOH. At the indicated times, 0.5 mL was removed and combined with 0.5 mL of 0.5 M ADH in 0.1 M HEPES, pH 8. After a 1 h reaction, each was desalted by a combination of dialysis and centrifugal filtration using an Amicon4 10 kDa cutoff device. The hydrazide and MenA concentrations were assayed to determine the hydrazide:MenA ratio.

**Figure 10 vaccines-08-00777-f010:**
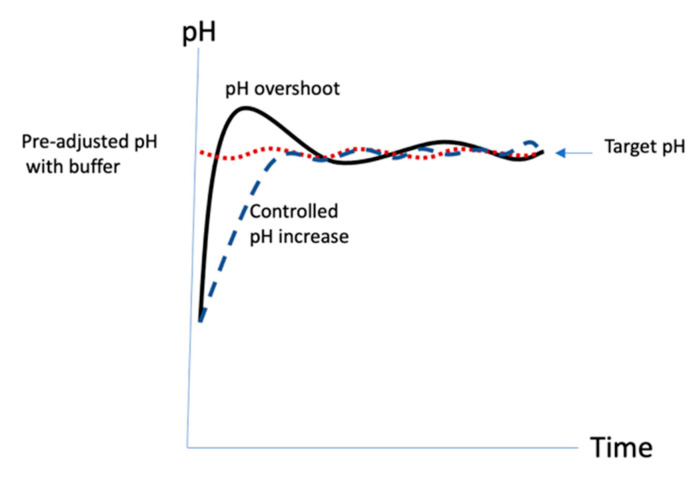
Modes of pH targeting for CDAP activation. Solid black line: original protocol with pH maintenance; blue dashed line: slow increase in pH using dilute base; red dotted line: Pre-adjustment of pH with DMAP.

**Figure 11 vaccines-08-00777-f011:**
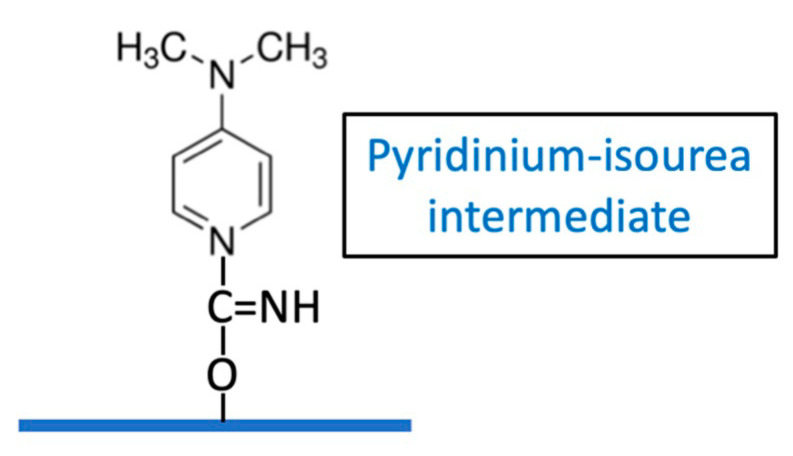
Proposed pyridinium–isourea structure as an intermediated in CDAP activated polysaccharides.

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
