# Peer review of "Activation of Soluble Polysaccharides with 1-Cyano-4-Dimethylaminopyridine Tetrafluoroborate (CDAP) for Use in Protein–Polysaccharide Conjugate Vaccines and Immunological Reagents. III Optimization of CDAP Activation"

_vaccines, 2020, doi:10.3390/vaccines8040777_

Round 1

Reviewer 1 Report

This is an important paper that addresses the optimization of CDAP reaction conditions used to activate polysaccharides for subsequent conjugation with proteins for vaccine construction.  Dr. Lees holds a number of patents utilizing CDAP as an activating agent for polysaccharides that are then conjugated to a protein (or other functional groups) for the production of a T-cell dependent immunogen.  This manuscript addresses the optimization of CDAP activating conditions that include pH, temperature, time, stability in various buffers and salts.  In addition, the manuscript addresses the issue of CDAP activation of polysaccharides that are base sensitive.  

The authors indicate that a future publication will address the optimization of the conjugation steps with CDAP-activated polysaccharides.

I have no major issues with the manuscript as written and have listed minor issues below that I feel might improve the manuscript.

Figure 2.- Consider labeling the left circle Activation and the right circle Conjugation?  Or A and B and indicating in the figure legend that the A circle is the CDAP activation step and the B circle the conjugation to protein step for clarity.

Lines 137-141.  The information about the UV spectra of CDAP (301nm) and DMAP (280nm) is already given in lines 192-193 in the Materials and Methods section.  If Figure 4 had the DMAP and CDAP peaks labeled it would not be necessary to repeat in the text.  Any idea what the peak at 260nm could be?

Line 262. enhanced rate of CDAP (of) consumption...  Delete (of).

Lines 278-279.  "We have found that the activated polysaccharide....than at room temperature (Lees, manuscript in preparation),...."  Doesn't Figure 6 in the current manuscript demonstrate the first part of this sentence and the rest of the sentence "allowing higher levels of conjugation when the process was carried out in the cold, while using less CDAP (Lees, manuscript in preparation)?

Line 378. " eliminated much of the variability (of) inherent in"  Delete (of)

Line 425.  "reactivity of polysaccharide hydroxyls would (be) reduced"  Add (be)

Line 445 in Figure 12 legend.  "derivatized dextran.  ADH (was) reacted with 2000kDa dextran was activated..."  Delete (was)

Line 540.  "can be used to pre-adjust the pH (of) the"  Add (of)

Line 547.  "CDAP hydrolysis (is) markedly slowed in the cold,..."  Add (is)

Line 552.  "The marked pH dependence (of) CDAP hydrolysis..."  Add (of)

Line 567.  Last word of line is misspelled.  amount

Line 576.  "high levels of polysaccharide activation can be achieved at (pH) 8 at 0..."  Add (pH)

Line 596.  "suggests that a pyridinium-isourea (may contribute it) cannot be the only..."  Delete (may contribute it)?  Not sure if this is what you mean to say.

Line 607  "so that only pH maintenance (is) necessary after adding CDAP."  Add is.

Author Response

I thank reviewer 1 for the helpful critique.  I have addressed all issues raised, as indicated below.  In addition, following suggestions from reviewer 2, I have moved figure 3 (pH change on addition of CDAP to water) and Figure 4 (UV spectra of CDAP and DMAP) to Supplement, along with Figure 7 (CDAP consumption in the presence and absence of Dextran) and Figure 10 (Stability of CDAP activated Dextran as function of pH).  Table 1 was also moved to the Supplement

Reviewer 1 comments

This is an important paper that addresses the optimization of CDAP reaction conditions used to activate polysaccharides for subsequent conjugation with proteins for vaccine construction.  Dr. Lees holds a number of patents utilizing CDAP as an activating agent for polysaccharides that are then conjugated to a protein (or other functional groups) for the production of a T-cell dependent immunogen.  This manuscript addresses the optimization of CDAP activating conditions that include pH, temperature, time, stability in various buffers and salts.  In addition, the manuscript addresses the issue of CDAP activation of polysaccharides that are base sensitive.  

The authors indicate that a future publication will address the optimization of the conjugation steps with CDAP-activated polysaccharides.

I have no major issues with the manuscript as written and have listed minor issues below that I feel might improve the manuscript.

Figure 2.- Consider labeling the left circle Activation and the right circle Conjugation?  Or A and B and indicating in the figure legend that the A circle is the CDAP activation step and the B circle the conjugation to protein step for clarity.

            Circles have been labeled Activation and Conjugation. Structures of CDAP, DMAP as well as the imidocarbonate and  carbonate derivatives added.

An additional Figure was added to provide a clear high level overview of CDAP activation without the side reactions.

Lines 137-141.  The information about the UV spectra of CDAP (301nm) and DMAP (280nm) is already given in lines 192-193 in the Materials and Methods section.  If Figure 4 had the DMAP and CDAP peaks labeled it would not be necessary to repeat in the text.  Any idea what the peak at 260nm could be?

The redundant sentence in lines 192-193 have been deleted.

A sentence has been added: In base, DMAP is not protonated and the peak shifts to 260 nm.

The figure was moved to Supplementary Materials.

Line 262. enhanced rate of CDAP (of) consumption...  Delete (of).

The offending duplicate (of) has been removed.

Lines 278-279.  "We have found that the activated polysaccharide....than at room temperature (Lees, manuscript in preparation),...."  Doesn't Figure 6 in the current manuscript demonstrate the first part of this sentence and the rest of the sentence "allowing higher levels of conjugation when the process was carried out in the cold, while using less CDAP (Lees, manuscript in preparation)?

Figure 6 refers to CDAP stability.  Lines 278-279 refer to the stability of the activated polysaccharide.  It would not necessarily be true that the activated polysaccharide would remain active for a significantly longer time.  As the conceptualization shown in Figure 2 indicates, the activated PS bridges the activation step and the conjugation step.   The beginning of the paragraph has been changed to:

In addition to CDAP being more stable in the cold, we have also found that the activated polysaccharide is more stable at 0°C than at room temperature (Lees, unpublished, manuscript in preparation), allowing higher levels of conjugation when the process was carried out in the cold.

Line 378. " eliminated much of the variability (of) inherent in"  Delete (of)

            This was fixed.

Line 425.  "reactivity of polysaccharide hydroxyls would (be) reduced"  Add (be)

            This was fixed

Line 445 in Figure 12 legend.  "derivatized dextran.  ADH (was) reacted with 2000kDa dextran was activated..."  Delete (was) This was fixed

Line 540.  "can be used to pre-adjust the pH (of) the"  Add (of). This was fixed

Line 547.  "CDAP hydrolysis (is) markedly slowed in the cold,..."  Add (is)  Changed to CDAP hydrolysis is markedly slower in the cold

Line 552.  "The marked pH dependence (of) CDAP hydrolysis..."  Add (of)  This was fixed

Line 567.  Last word of line is misspelled.  Amount  This was fixed

Line 576.  "high levels of polysaccharide activation can be achieved at (pH) 8 at 0..."  Add (pH)  This was fixed

Line 596.  "suggests that a pyridinium-isourea (may contribute it) cannot be the only..."  Delete (may contribute it)?  Not sure if this is what you mean to say.

This paragraph has been simplified and revised.  It should now be more clear.

The structure of the pyridinium isourea intermediate has been added as Figure 14.

Line 607  "so that only pH maintenance (is) necessary after adding CDAP."  Add is.  Fixed

Reviewer 2 Report

The manuscript focuses on CDAP chemistry as it relates to polysaccharide activation for glycoconjugate vaccines. This work seeks to further delineate the stability of CDAP (how much remains in the active vs. the inactive hydrolyzed form) as it relates to pH, temperature and time. These attributes directly affect the amount of active CDAP that is present to activate polysaccharides for conjugation to a carrier protein. The authors present a lot of data that suggest that proper regulation of pH at lower temperature conditions is necessary to preserve stability of CDAP. I commend the authors for the very thorough investigation using an array of analytical techniques. The paper could be shortened by keeping only the most key figures perhaps placing some figures into a supplemental section. In addition, this paper would be improved by better visual representations. Specifically:

  • More chemical structures of the molecules described would better help the reader follow along with the chemical changes as it relates to the data (In Fig. 2: structure of carbamate and imidocarbonate, structural representation of -OCN; Fig. 4 addition of CDAP structure and DMAP structure; In Table 1, structure of chemicals included
  • A summary table of the major findings of their investigations at the end of the results
  • A checklist of information for researchers to be mindful of when using CDAP chemistry to activate polysaccharides to accompany section 4.1 A 

Other notes: Fig. 9 which shows SEC to affirm conjugate formation is missing from the manuscript, although this Figure referenced.

Author Response

 I thank Reviewer 2 for their helpful comments. 

I took your suggestion to move material to a supplement:

I have moved figure 3 (pH change on addition of CDAP to water) and Figure 4 (UV spectra of CDAP and DMAP) to Supplement, along with Figure 7 (CDAP consumption in the presence and absence of Dextran) and Figure 10 (Stability of CDAP activated Dextran as function of pH).  Table 1 was also moved to the Supplement.

I simplified the discussion of the likely active intermediate and added a figure with the structure of the proposed pyridinium isourea intermediate.

Reviewer 2. The manuscript focuses on CDAP chemistry as it relates to polysaccharide activation for glycoconjugate vaccines. This work seeks to further delineate the stability of CDAP (how much remains in the active vs. the inactive hydrolyzed form) as it relates to pH, temperature and time. These attributes directly affect the amount of active CDAP that is present to activate polysaccharides for conjugation to a carrier protein. The authors present a lot of data that suggest that proper regulation of pH at lower temperature conditions is necessary to preserve stability of CDAP. I commend the authors for the very thorough investigation using an array of analytical techniques. The paper could be shortened by keeping only the most key figures perhaps placing some figures into a supplemental section. In addition, this paper would be improved by better visual representations. Specifically:

  • More chemical structures of the molecules described would better help the reader follow along with the chemical changes as it relates to the data (In Fig. 2: structure of carbamate and imidocarbonate, structural representation of -OCN;

Fig. 4 addition of CDAP structure and DMAP structure as well as the stuctures of the side reactions.

In Table 1, structure of chemicals included (now in Supplement)

  • A summary table of the major findings of their investigations at the end of the results.
    • I added a box with the major findings to the end of the discussion.
  • A checklist of information for researchers to be mindful of when using CDAP chemistry to activate polysaccharides to accompany section 4.1 A 

A Figure has been added to show the various modes of adjusting the pH for CDAP activation. This should clarify the method of using DMAP to pre-adjust the pH prior to the addition of CDAP, as well as to emphasize the importance of slowly raising the pH.

Other notes: Fig. 9 which shows SEC to affirm conjugate formation is missing from the manuscript, although this Figure referenced.

            The omitted Figure 9 has been added.